# Acetylcholine and Royal Jelly Fatty Acid Combinations as Potential Dry Eye Treatment Components in Mice

**DOI:** 10.3390/nu13082536

**Published:** 2021-07-24

**Authors:** Masayuki Yamaga, Toshihiro Imada, Hiroko Tani, Shigeru Nakamura, Ayanori Yamaki, Kazuo Tsubota

**Affiliations:** 1Institute for Bee Products and Health Science, Yamada Bee Company, Inc., Okayama 708-0393, Japan; my1636@yamada-bee.com (M.Y.); ht0807@yamada-bee.com (H.T.); ay1255@yamada-bee.com (A.Y.); 2Department of Ophthalmology, School of Medicine, Keio University, Tokyo 160-8582, Japan; imada_toshihiro@keio.jp (T.I.); s-nakamura.a5@keio.jp (S.N.); 3Tsubota Laboratory, Inc., Shinjuku-ku, Tokyo 160-0016, Japan

**Keywords:** royal jelly, acetylcholine, fatty acid, ophthalmology, dry eye

## Abstract

Dry eye is a multifactorial disease characterized by ocular discomfort and visual impairment. Our previous studies have shown that royal jelly (RJ) has restored the capacity for tear secretion by modulating muscarinic calcium signaling. RJ contains acetylcholine, which is a major cholinergic neurotransmitter, and a unique set of fatty acids with C 8 to 12 chains, which are expected to be associated with health benefits. The purpose of the present study was to investigate the active components involved in tear secretion capacity, focusing on acetylcholine and fatty acids in RJ. Using the stress-induced dry-eye model mice, it was confirmed that acetylcholine with three fatty acids (10-hydroxydecanoic acid, 8-hydroxyoctanoic acid, and (*R*)-3,10-dihydroxydecanoic acid) was essential for tear secretion. In ex vivo Ca^2+^ imaging, these three fatty acids suppressed the decrease in intracellular modulation of Ca^2+^ in the lacrimal gland by acetylcholine when treated with acetylcholinesterase, indicating that the specific type of RJ fatty acids contributed to the stability of acetylcholine. To our knowledge, this study is the first to confirm that a specific compound combination is important for the pharmacological activities of RJ. Our results elucidate the active molecules and efficacy mechanisms of RJ.

## 1. Introduction

Royal jelly (RJ) is a yellowish white substance, a cream secreted by the cephalic glands of honey bees (*Apis mellifera* L.) and used as nutrition for young larvae and queen bees. RJ is known to induce epigenetic development in the queen bee and is considered to be involved in many advantages for the queen bee, including enabling a large size (double that of worker bees), fertility, and a long lifespan (about 5–6 years, compared to worker bees with a lifespan of 35–40 days) [1]. RJ has a multitude of pharmacological activities reported in clinical trials, including antihypertensive effects [2]; the alleviation of chills [3], neck muscle strain [4], and female menopausal symptoms [5]; reduction of blood sugar in patients with type 2 diabetes [6]; prevention of age-associated muscle strength decline [7]; and maintenance of skin moisture [8]. RJ consists of water (60–70%), proteins (9–18%), sugar (7.5%), lipids (3–8%), and other trace compounds. (*E*)-10-Hydroxy-2-decenoic acid (10H2DA) and 10-hydroxydecanoic acid (10HDAA) are known to be major fatty acids in RJ and represent 60–80% of RJ lipids [9]. These fatty acids have been reported to activate TRPA1 channels expressed in HEK293 cells [10], induce estrogen receptor β recruitment to the promoter in MCF-7 cells [11], express GLUT4 in skeletal muscle by activating AMPK signaling in vivo [12], and enhance filaggrin production in a human three-dimensional epidermis model [13]. Moreover, the composition of RJ fatty acids has been characterized as a set of C8, C10, and C12 fatty acids [14]. In the case of minor fatty acids, several pharmacological activities have been reported, such as anticancer [15], antimicrobial [16], and anti-inflammatory [17] activities. Therefore, RJ fatty acids are expected to play an important role in the pharmacological activities of RJ reported in clinical trials.

Dry eye is a multifactorial disease characterized by an unstable tear film that leads to a variety of symptoms and potentially even visual impairment and ocular surface damage [18]. The incidence of dry eye has increased because of the use of air conditioning and digital devices, and the prevalence of dry-eye disease has been reported to range from 5 to 50% [19]. Current treatments are generally the instillation of artificial tears that can provide temporary relief, but the effect of topical treatments is not persistent because it does not solve the underlying causes [20].

Previously, our research group conducted a double-blind, placebo-controlled trial to investigate the effects of RJ on dry eye signs and symptoms in human patients. In this study, we found that supplementation with oral RJ for 8 weeks can improve tear secretion, as assessed by the Schirmer score, in patients with dry-eye symptoms [21]. In addition, we investigated the underlying lacrimal mechanisms in relation to the results of this study using a rodent model of dry eye. A rat blink-suppressed dry-eye model, simulating the effect on dry eye in patients whose etiology is associated with excessive staring at a computer display, which corresponds to a human trial, has been investigated. It has been assumed that these effects occur alongside an increase in ATP, mitochondrial function, and phosphorylation of AMPK by modulating the muscarinic calcium-signaling pathway stimulated by RJ, reflecting the restoration of the energy state of the lacrimal gland (LG) [22]. Given that acetylcholine (ACh), a major cholinergic neurotransmitter involved in tear secretion, has been shown to be 1 mg/g in RJ [23], ACh may be a potential RJ component for restoring tear secretion. In the present study, in order to investigate a potential RJ component for the treatment of dry eye, we evaluated the effect of RJ components on tear secretion capacity using a rodent dry-eye model focused on RJ constitutive fatty acids and ACh.

## 2. Materials and Methods

### 2.1. Materials and Chemicals

Enzyme-treated RJ powder (ETRJ), which degrades RJ proteins to peptides and amino acids by proteases to reduce the immunoreactivity and allergenicity [24], was obtained from Yamada Bee Company, Inc. (Okayama, Japan), and standardized to contain a minimum of 3.5% 10H2DA and 0.6% 10HDAA. Table 1 shows the content of ETRJ compounds and the source or reference of each compound used in this study. The synthetic method and analytical data are provided in Supporting Information (Appendix A).

### 2.2. Animals

All animal experiments in this study were approved by the Ethics Committee for Animal Research at the Keio University School of Medicine (approval no. 11008). All mice were treated according to the Association of Research and Vision in Ophthalmology (ARVO) statement for the Use of Animals in Ophthalmic Vision Research.

Female C57BL/6 mice (Charles River, Yokohama, Japan) and yellow cameleon-Nano15 (YCNano15) transgenic mice were used in this study (8 weeks old, 20–23 g). YCNano15 is a mouse line expressing the YCNano15 Ca^2+^ sensor probe under the control of the CAG promoter [29]. They were quarantined and acclimatized for 1 week prior to the experiments under the following general conditions: room temperature of 23 ± 2 °C, relative humidity of 60 ± 10%, alternating 12 h light-dark cycle (8 a.m.–8 p.m.), water, and food ad libitum.

### 2.3. Stress-Induced Dry-Eye Model in Mice

We used a mouse stress-induced dry-eye model that simulates evaporative dry eye in patients whose etiology is associated with excessive staring at a computer display [30]. In brief, the mice were restrained in a 50 mL plastic conical tube and treated with a flow of air directed toward their heads at a rate of 0.5–1.0 m/s for 4 h. They were placed individually in cages, with water and food available ad libitum for the remaining time. ETRJ was suspended in distilled water, and a dose of 300 mg/kg was administered orally using a feeding needle (20 gauge) before stress exposure. To determine the active component of RJ, oral administration of ACh and each fatty acid corresponding to the content of 300 mg/kg ETRJ (Table 1) was performed individually or in combination with distilled water. Distilled water was used as a vehicle control. Six mice were used in each experiment.

### 2.4. Post-Ganglionic Denervation of the Lacrimal Gland (PGD)

PGD was performed as described previously [31]. In brief, the mice were placed in a prone position, and the skin on the temporal side of the head was incised under deep anesthesia using a combination of medetomidine (0.75 mg/kg), butorphanol (5 mg/kg), and midazolam (4 mg/kg). The post-ganglionic nerve bundle was detached from the blood vessels at the caudal root site of the ventral surface of the LG and denervated under a stereomicroscope. PGD was performed unilaterally. They were placed individually in cages, with water and food available ad libitum during the experimental period. Three hundred mg/kg ETRJ, ACh, three selected fatty acids (10HDAA, 8HOA, and 3,10DDA) mixture, and a combination of ACh with a mixture of three fatty acids were dissolved in distilled water and administered orally using a feeding needle (20 gauge). Distilled water was used as a vehicle control. The dose of ACh and each of the three selected fatty acids corresponded to the content of 300 mg/kg ETRJ. Each solution was administered orally twice daily after PGD surgery. In the sham group, distilled water was administered orally twice daily. Five mice were used in each experiment.

### 2.5. Measurement of Tear Secretion in Mice

Tear secretion was measured using a modified phenol red thread test with a phenol red thread (Zone-Quick; Showa Yakuhin Kako, Tokyo, Japan) [30]. It was placed on the temporal side of the conjunctiva between the limbus and outer canthus for 15 s. The length of the moistened area at the edge was measured to be within 0.5 mm. For the stress-induced dry-eye model in mice, tear secretion was measured before and 1 d after stress exposure. The average values of both the left and right eyes were used for the analysis. The percentage of tear secretion value at 1 d after stress exposure relative to that before stress exposure was calculated. For PGD mice, tear secretion was measured before and 1, 2, 3, 5, and 7 days after PGD in the PDG-treated unilateral eye. The values of the PGD-treated eyes were used in the analysis.

### 2.6. Histopathological Analysis

PGD mice were euthanized with an overdose of sodium pentobarbital 7 days after PGD surgery and their LGs were dissected. The entire LG was fixed in 10% formalin solution and embedded in paraffin. Sections (5 mm thick) were obtained from the middle of the horizontal dissection of the LG. The sections were then stained with hematoxylin and eosin. The images were captured using a BIOREVO BZ-9000 (Keyence, Osaka, Japan) optical microscope. To quantify acinar cell size, the size of 20 acinar cells was measured in three randomly selected areas in each section using the BZ-Analyzer 2.1 software (Keyence, Osaka, Japan).

### 2.7. Acetylcholinesterase Treatment

ETRJ (500 mg), the corresponding amounts of acetylcholine, and three selected fatty acids (10HDAA, 8HOA, and 3,10DDA) were suspended in 10 mL of saline solution (140 mM NaCl, 5 mM KCl, 1 mM MgCl_2_, 10 mM HEPES, and 10 mM dextrose [pH7.4]). These solutions were incubated with 0.08 U/mL of acetylcholinesterase (AChE) at 37 °C for 30 min. After incubation, rivastigmine, an AChE inhibitor, was added at a final concentration of 100 µM to stop the reaction. These mixtures, before and after AChE treatment, were evaluated by intracellular Ca^2+^ imaging, and the acetylcholine in these samples was quantified by LC/MS/MS.

### 2.8. Ex Vivo Ca^2+^ Imaging in the Lacrimal Gland

YCNano15 transgenic mice were euthanized with an overdose of sodium pentobarbital. The LGs dissected from YCNano15 mice were transferred to round coverslips that were mounted on the bath region of a perfusion chamber and continuously perfused with a saline solution through polyethylene tubes connected to a Masterflex peristaltic pump (Cole-Parmer, Chicago, IL, USA) at a flow rate of 0.8 mL per minutes. The mixture of ETRJ (500 µg/mL), ACh (0.115 µg/mL), and the three fatty acids (10HDAA: 6.735 µg/mL, 8HOA: 1.560 µg/mL, and 3,10DDA: 1.815 µg/mL) was diluted to the desired concentration with a saline solution before use. Each stimulant that reacted with and without AChE was applied to the LG for 1 min at 5 min intervals.

Ca^2+^ imaging was performed in accordance with previous reports [29]. Briefly, Ca^2+^ mobilization was observed with a two-photon microscope equipped with a 25× water-immersion objective lens by measuring the change in the fluorescence resonance energy transfer (FRET) ratio, which was calculated as the ratio of YFP to CFP fluorescence intensity. An excitation wavelength of 830 nm was used for FRET imaging. Two-photon excited fluorescence images of CFP and YFP were acquired in separate channels through dichroic mirrors and emission filters, namely BP460-500 nm and BP520-560 nm, respectively. The fluorescence images were acquired at approximately 1 frame/s for 13 min at 2 µm depth-intervals from the LG surface at a maximum depth of 10 µm. Fluorescence images were reconstructed using Imaris software and analyzed using MATLAB software (MathWorks, Natick, MA, USA).

### 2.9. Analysis of Acetylcholine and RJ Fatty Acids by LC/MS/MS

LC/MS/MS was performed on a UPLC system (Ultimate 3000, Thermo Scientific, Waltham, MA, USA) with an MS orbitrap system (Q-Exactive Focus, Thermo Scientific). 

ACh contained in ETRJ was extracted with 20% methanol and diluted to 100 μg/mL. Acetylcholine was analyzed using a normal phase column (Atlantis Hilic Silica, 2.1 × 50 mm, 3 µm i.d., Waters, Milford, MA, USA) at 40 °C. The mobile phase consisting of water with 0.1% formic acid (A) and acetonitrile (B) was pumped at a flow rate of 0.3 mL/min. The gradient system was as follows: 95% B (0–2 min), 95–20% B (2–8 min), 20% B (8–12 min), and 95% B (12–20 min). The typical injection volume was 3 µL. RJ fatty acid analysis was performed according to a previous report [28]. Briefly, RJ fatty acids contained in ETRJ were extracted with 100% methanol and diluted to 10 or 100 μg/mL. The sample was injected onto a reversed-phase column (Acquity UPLC BEH C18, 2.1 × 100 mm, 1.7 µm i.d., Waters) and eluted using a gradient with solvent A, 0.01% acetic acid and solvent B, acetonitrile at a flow rate of 0.3 mL/min at 40 °C. The gradient system was as follows: 5% B (0–2 min), 5–100% B (2–17 min), 100% B (17–20 min), and 5% B (20–25 min). MS/MS was performed in positive and negative modes with full mass monitoring (range *m*/*z* 70–1000). MS conditions were the same as those described in our previous report [28]. The validation data of the analysis methods are provided in the Supporting Information (Appendix A).

### 2.10. Statistical Analysis

All results are presented as mean ± SEM, and statistical analyses were performed using JMP12 software (version 12.2; SAS Institute, Cary, NC, USA). Comparisons between the two groups were performed using an F-test, followed by a *t*-test for parametric variables and a U-test for nonparametric variables. Multiple comparisons were performed using a one-way analysis of variance, followed by the Tukey-Kramer or Dunnett test. Differences between the measurement variables were considered significant if the resulting *p*-value was ≤0.05.

## 3. Results

### 3.1. ACh and RJ Fatty Acids Are Necessary Components of Dry-Eye Suppression in Royal Jelly

Initially, we orally administered ACh or a mixture of nine RJ fatty acids (nine RJ FAs) in a stress-induced dry-eye mouse model. The effects of ETRJ, ACh, and nine RJ FAs are shown in Figure 1A. ETRJ and significantly preserved tear secretion (*p* < 0.001 vs. vehicle), and ACh and nine RJ FAs did not affect the decreased tear secretion compared to the vehicle.

We then evaluated the combined effects of ACh and nine RJ FAs or each fatty acid. The ameliorative effect of tear secretion was not observed in ACh treated with RJ fatty acids. Interestingly, significant restoration of tear secretion was observed in ACh with nine RJ FAs compared to the vehicle (Figure 1B). Typical changes in the tear secretory patterns are shown in Figure 1C. These results show that ACh with two or more RJ fatty acids is essential to restore decreased tear secretion. To select a suitable combination of RJ fatty acids, nine RJ FAs were grouped into saturated C8-10 chain fatty acids (SC8-10 FAs; 10HDAA, 8HOA, 3,10DDA, and SA), and the other fatty acids (10H2DA, 2DA, 9,10D2DA, 11,12D2DA, and 12HDA). The mixture of ACh with SC8-10 FAs restored the reduction in tear secretion to the same level as that of 9 RJ FAs. In contrast, the mixture of other FAs did not restore reduced tear secretion (Figure 2A).

Next, to identify potent combinations of fatty acids, four SC8-10 FAs were grouped into five mixtures composed of two or three SC8-10 FAs (Figure 2B). The ameliorating effects of ACh with these SC8-10 FA mixtures on the reduction of tear secretion were compared. For ACh either with 10HDAA/8HOA, 8HOA/3,10DDA, or 10HDAA/3,10DDA, the reduction in tear secretion was at the same level as that of the vehicle. For ACh with a mixture of 10HDAA, 8HOA, and 3,10DDA (10HDAA/8HOA/3,10DDA), tear secretion was restored to the same level as ACh with a mixture of four SC8-10 FAs (Figure 2B). These results suggest that ACh with 10HDAA/8HOA/3,10DDA is an essential component for the anti-dry-eye effect of RJ.

### 3.2. ACh with 10HDAA/8HOA/3,10DDA Preserved Tear Secretion Capacity and LG Morphology in the LG Post-Ganglionic Denervation Dry-Eye Model

Our previous study demonstrated that the oral administration of RJ restores tear secretion and LG structure when neuronal LG stimuli are interrupted [21]. These findings indicate that direct effects on LGs are a potential mechanism of RJ supplementation in patients with dry eye. To further confirm the underlying mechanism of ACh with 10HDAA/8HOA/3,10DDA on the restoration of tear secretion similar to that of RJ, we compared changes in body weight, tear secretion, and LG pathology between six groups: (1) sham, (2) vehicle, (3) ETRJ, (4) ACh, (5) 10HDAA/8HOA/3,10DDA, and (6) ACh with 10HDAA/8HOA/3,10DDA. The evaluation was performed when neuronal stimuli from the autonomic central nervous system to the LG were interrupted in the PGD dry-eye model.

There was no change in body weight between the groups (Figure 3A). In the vehicle group, tear secretion significantly decreased immediately after denervation, and this reduction was sustained until day 7. In the ETRJ group, significant preservation of tear secretion was observed compared to that in the vehicle group during the experimental period. The values were approximately 80% before PGD. In the ACh and 10HDAA/8HOA/3,10DDA group, tear secretion was at the same level as that in the vehicle group. For the ACh with 10HDAA/8HOA/3,10DDA group, significant preservation of tear secretion was observed compared to the vehicle group. The values and patterns were similar to those in the ETRJ group. The sham eye did not affect tear secretion (Figure 3B). Since decreased tear secretion and LG atrophy were characteristic changes in this dry eye model, gross pathological and histological changes were evaluated. Typical changes among these groups are shown in Figure 3C on the left. PGD-induced LG atrophy, reduced organ size (Figure 3C, upper panels), and acinar cell area (Figure 3C, lower panels), were significantly suppressed in the ETRJ and ACh with 10HDAA/8HOA/3,10DDA groups (Figure 3C, right).

### 3.3. 10HDAA/8HOA/3,10DDA Suppressed the Decrease of ACh-Modulated [Ca^2+^]i in the LG by Acetylcholinesterase Treatment

ACh is rapidly hydrolyzed and diminishes the physiological activity of acetylcholinesterase (AChE) at brain cholinergic synapses and neuromuscular junctions [32]. To investigate whether 10HDAA/8HOA/3,10DDA preserves the activity of ACh from AChE decomposition, we evaluated the effect of 10HDAA/8HOA/3,10DDA on Ach-modulated LG [Ca^2+^]i changes. Modulation of LG [Ca^2+^]i by ACh was diminished after treatment with AChE (Figure 4A). Modulation of LG [Ca^2+^]i by ETRJ (Figure 4B) and ACh with 10HDAA/8HOA/3,10DDA were preserved after AChE treatment (Figure 4C). Significant preservation of LG [Ca^2+^]i changes was observed in ACh with 10HDAA/8HOA/3,10DDA compared to ACh. The LG [Ca^2+^]i changes were preserved at the same level as those induced by ETRJ (Figure 4D). Corresponding to the response to LG [Ca^2+^]i changes, the analytical quantification of ACh by LCMS showed that 10HDAA/8HOA/3,10DDA preserved the amount of ACh from AChE (Figure 4E).

## 4. Discussion

In the present study, we found that three RJ fatty acids consisting of C8-10 chains, 10HDAA/8HOA/3,10DDA, in combination with ACh, are the RJ components that may have potential application in the treatment of dry eye.

ACh is an important neurotransmitter that is synthesized in preganglionic neurons in both sympathetic and parasympathetic autonomic ganglia. ACh is released at cholinergic synaptic sites in response to neural excitability. The released ACh is rapidly hydrolyzed by AChE and decreases physiological activity at brain cholinergic synapses and neuromuscular junctions. Furthermore, it has been reported that AChE and butyrylcholinesterase were expressed in the blood and intestine epithelial cells [33,34]. Therefore, it is assumed that orally administered ACh is mostly degenerated in the process of absorption and blood transport before it reaches the ACh receptor expressed in brain cholinergic synapses and autonomously innervated organs, such as smooth muscles of blood vessels and intestines, pancreas, and LGs. Nevertheless, it has been reported that orally administered RJ has ACh-like effects, such as improvement of Alzheimer’s disease [35], vasodilation induced by nitric oxide production [36], and induction of insulin secretion [37]. Moreover, our previous report showed that RJ had a tear secretion capability via muscarinic acetylcholine receptor (mAChR) signaling induced by intracellular Ca^2+^ increase in LGs, which was suppressed by the mAChR antagonist atropine and by inhibitors of phospholipase C and ER-Ca^2+^-ATPase essential enzyme pathways [23]. These reports and our results (Figure 3 and Figure 4) indicate that ACh in RJ retains its activity even after oral administration of RJ, possibly due to the effect of RJ fatty acids, preventing ACh degradation by AChE and transporting ACh to targeted sites efficiently. Additionally, decanoic acid (capric acid) has been reported to interact with mAChR agonists in the ileum and jejunum [38]. Thus, 10HDAA, 8HOA, and 3,10DDA, which are saturated medium-chain fatty acids similar to capric acid, could interact with mAChR. In the nicotinic ACh receptor (nAChR), lipids have been reported to be potent modulators that influence receptor function both by conformational selection and by kinetic mechanisms via membrane stabilization [39]. In addition, the binding affinity of mAChR and its substrates has been reported to change depending on the membrane lipid composition [40]. 10HDAA, 8HOA, and 3,10DDA could play the role of mAChR modulators to stabilize lipid membranes and/or binding to allosteric sites. Further studies are needed to calculate the molecular modeling of the mAChR-fatty acid interactions to clarify the mechanism of RJ tear secretion capacity.

## 5. Conclusions

For the first time, we identified RJ components for the treatment of dry eye. Our results suggest that the combination of ACh with three saturated fatty acids, namely 10HDAA, 8HOA, and 3,10DDA, which have a terminal hydroxyl group, was critical for the therapeutic effect of RJ in dry eye. The restoration of Ach content using these three fatty acids may function by inhibition of AChE activity as a possible mechanism.

## Figures and Tables

**Figure 1 nutrients-13-02536-f001:**
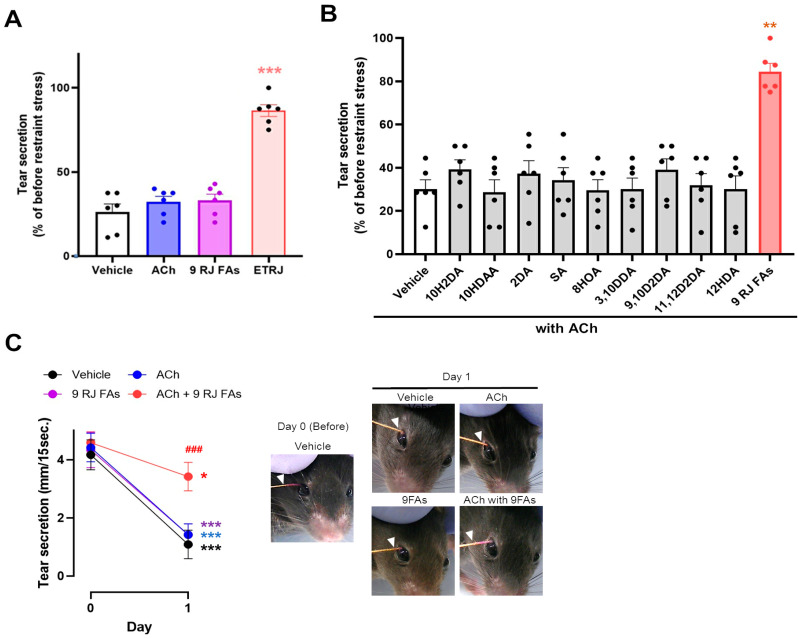
ACh with two or more RJ fatty acids were essential to restore decreased tear secretion. (**A**) Effect of ACh and 9 RJ fatty acids (9 RJ FAs). (**B**) Effect of a combination of ACh with RJ fatty acids on tear secretion. (**C**) Change in measured value in tear secretion (**left**) and representative photographs measured with a cotton thread (**right**). The value was calculated as a percentage of tear secretion at 1 day after restraint stress exposure compared to before restraint stress. (**A**,**B**) All data represent the mean ± SEM, *n* = 6 mice eyes. * *p* < 0.05, ** *p* < 0.01, *** *p* < 0.001 versus vehicle (**A**,**B**) or day 0, ### *p* < 0.001 versus vehicle (**C**).

**Figure 2 nutrients-13-02536-f002:**
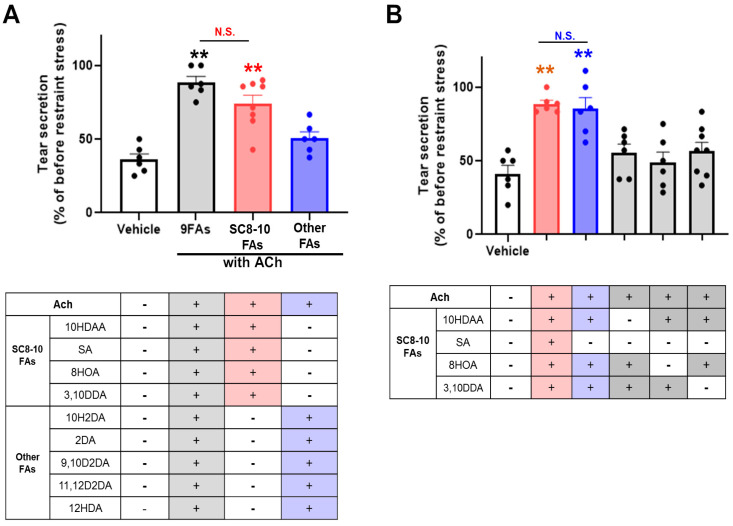
ACh with 10HDAA/8HOA/3,10DDA were the essential components for the anti-dry-eye effect of RJ. (**A**) Effect of saturated C8-10 chains fatty acids (SC8-10 FAs) and the other fatty acids (other FAs) on tear secretion (upper). (**B**) Effect of SC8-10 FAs on tear secretion. All data represent the mean ± SEM, *n* = 6 mice eyes. ** *p* < 0.01 versus vehicle. N.S. indicates no significant difference.

**Figure 3 nutrients-13-02536-f003:**
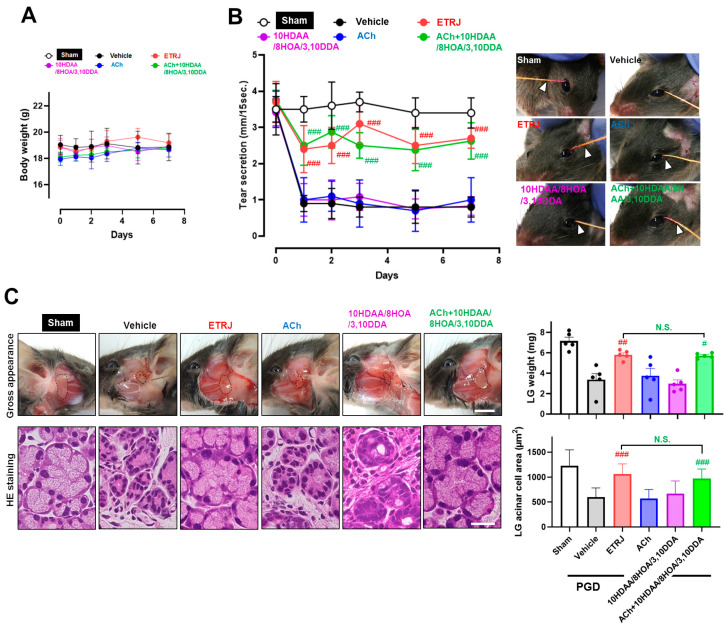
The mechanism of tear secretion restoration by ACh with 10HDAA/8HOA/3,10DDA is similar to that of RJ. Changes in (**A**) body weight and (**B**) tear secretion. The photographs on the right side show representative tear secretion patterns measured with a cotton thread 7 days after post-ganglionic denervation (PGD) surgery. (**B**) The arrow indicates the wetted length due to tear secretion. (**C**) Gross (upper) and histological (lower) changes in the LG. The upper and lower right bar chart show the LG weight and LG acinar cell area, respectively. The scale bar is 5 mm (upper) and 20 µm (lower). All data represent the mean ± SEM, *n* = 5 mice. # *p* < 0.05, ## *p* < 0.01, ### *p* < 0.001 versus vehicle. N.S. indicates the no significant difference between ETRJ and ACh + 10HDAA/8HOA/3,10DDA.

**Figure 4 nutrients-13-02536-f004:**
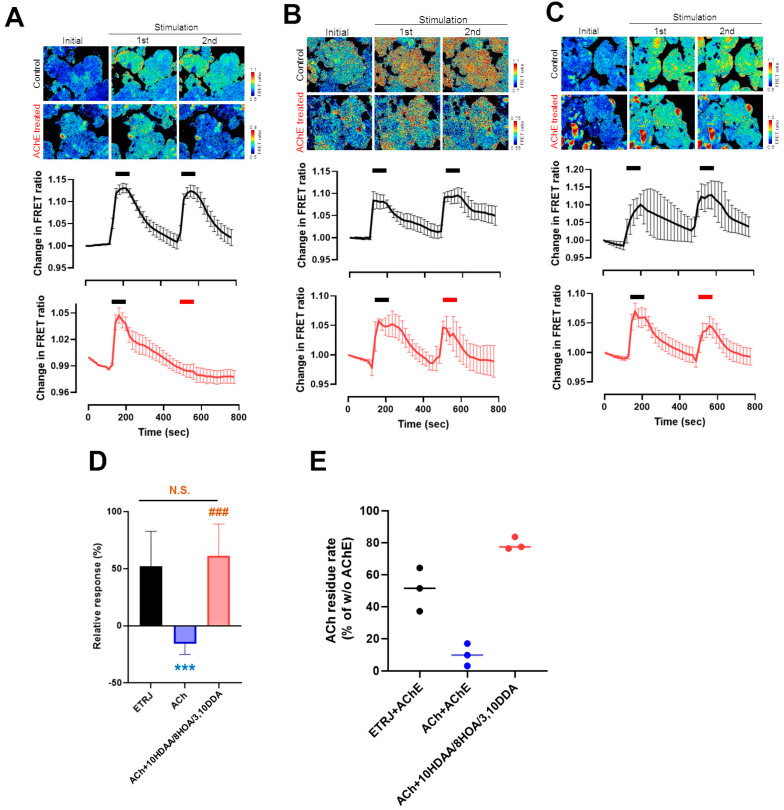
10HDAA/8HOA/3,10DDA suppressed the decrease of Ach-modulated LG [Ca^2+^]i by ACh esterase. [Ca^2+^]i increase in LG acinar cells exposed to (**A**) ACh, (**B**) ETRJ, and (**C**) ACh + 10HDAA/8HOA/3,10DDA treated with or without ACh esterase (AChE). The upper panels show pseudo-colored images of [Ca^2+^]i. The lower trace shows the [Ca^2+^]i response to each stimulus. The black and red bar over the trace indicates when each stimulus reacted without and with AChE, respectively. (**D**) Summarized data on amplitudes of [Ca^2+^]i responses. Relative responses were calculated as a percentage of stimulus reacted with the AChE induced [Ca^2+^]i response relative to the stimulation induced without AChE. (**E**) Analytical quantification of ACh. All data represent the mean ± SEM, *n* = 4–10 LGs (**A**–**D**) and *n* = 3 (**E**). *** *p* < 0.001 versus ETRJ, ### *p* < 0.001 versus ACh. N.S. indicates no significant difference between ETRJ and ACh with 10HDAA/8HOA/3,10DDA (**D**).

**Table 1 nutrients-13-02536-t001:** The composition of the enzyme-treated royal jelly (ETRJ) and source or reference of the compounds used in this study.

Compounds	(%) ^1^	Oral Administration Dosage (mg/kg) ^2^	Source or Reference
acetylcholine (ACh)	0.023	0.069	Tokyo Chemical (Tokyo, Japan)
8-hydroxyoctanoic acid (8HOA)	0.312	0.936	Sigma-Aldrich (St. Louis, MO, USA)
(*R*)-3,10-dihydroxydecanoic acid (3,10DDA)	0.363	1.089	Isolation from RJ according to the previous report (Noda et al. [25]).
10-hydroxydecanoic acid (10HDAA)	1.347	4.041	Combi-Blocks (San Diego, CA, USA)
(*E*)-9,10-dihydroxy-2-decenoic acid (9,10D2DA)	0.001	0.003	Prepared by the mixing of synthetic (*R*) and (*S*)-acids in a ratio of *R*/*S* = 3.5/1 (Tani et al. [26])
(*E*)-10-hydroxy-2-decenoic acid (10H2DA)	4.267	12.801	Hangzhou Eastbiopharm (Hangzhou, China)
(*E*)-2-decenedioic acid (2DA)	0.435	1.305	Sundia MediTech (Shanghai, China)
sebacic acid (SA)	0.279	0.837	Sigma-Aldrich (St. Louis, MO, USA)
(*E*,*R*)-11,12-dihydroxy-2-dodecenoic acid (11,12D2DA)	0.001	0.003	Prepared according to the procedures described in the patent [27] and Supporting Information (Scheme S1). Analytical data are provided in the Supporting Information (Appendix A).
12-hydroxydodecanoic acid (12HDA)	0.049	0.147	MP Biomedicals (Santa Ana, CA, USA)

^1^ ACh, 9,10D2DA, and 11,12D2DA content in ETRJ were analyzed by LC/MS/MS, and the other fatty acid content in ETRJ was based on data from a previous report [28]. ^2^ Equivalent to the content in 300 mg/kg ETRJ.

## Data Availability

The datasets generated and/or analyzed during the current study are available from the corresponding author on reasonable request.

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
