# Peer review of "Acetylcholine and Royal Jelly Fatty Acid Combinations as Potential Dry Eye Treatment Components in Mice"

_nutrients, 2021, doi:10.3390/nu13082536_

Round 1

Reviewer 1 Report

In this experimental study, the authors identified the potentially effective components of orally administered royal jelly (RJ) in stimulating lacrimal gland (LG) secretion using a mouse model of dry eye disease (increased tear evaporation). The in vivo results showed that the combination of acetylcholine (ACh) and 3 fatty acids (10HDAA/8HOA/3,10DDA) were responsible for the increased LG size and tear production. In vitro experiments suggested that the selected fatty acids in RJ had inhibitory effects on acetylcholinesterase. This study was clearly presented, and the manuscript was well-written. However, there are some important points to be addressed before being considered for publication.

Major points:

  1. As dry eye disease is a heterogeneous population, the authors should clarify that the selected mouse model was used to model the dry eye mainly due to increased tear evaporation (i.e., excessive staring at a computer display). Therefore, this model was not developed for other common causes of dry eye disease, such as insufficient tear production, abnormal tears, inflammation, or a combination of these factors.
  2. The authors should clarify how data from two eyes of the same animal were treated in the statistical analysis. These data of the same animal were correlated and were not independent. Therefore, they should not be treated as independent samples in all the mean-based statistics. This affected many results of descriptive analyses and comparative analyses.
  3. Please provide direct evidence supporting this statement in conclusion - '...includes direct inhibition of AChE', such as measuring the level of activities of acetylcholinesterase with and without the three fatty acids.

Minor points

  1. The authors would want to avoid emphasizing the randomized controlled clinical trial (PLoS One 2017, 12(1), e0169069). As the study groups were imbalanced in age and gender (a marginal P-value), differences in study outcomes were to be confirmed in future studies. However, the increased tear secretion was demonstrated in patients with dry-eye symptoms.
  2. "...which corresponds to human trail has been performed" should be "...which corresponds to human trial has been performed."
  3. Please describe the definitions of the error bars in each plot.

Reviewer 2 Report

The authors have presented a well-written and scientifically sound study.

They present interesting data on the constituents of Royal Jelly (RJ) that have a beneficial effect in dry eye disease and suggested a probable mechanism.

The reviewer has some minor suggestions, as follows:

  1. The manuscript needs a thorough evaluation of English and grammar, especially on the writing style. There are some instances when the sentence is grammatically sound but the writing style changes the meaning. For example:
  • Page 2, line 10: it is though to be a “tear film stabilizing product” should be rewritten as “Dry-eye is a multifactorial disease of the ocular surface, resulting from deficiencies in volume and/or evaporation of the tear film [18], which in turn can lead to tear film destabilization causing surface damage, inflammation, discomfort, and visual impairment [19].
  • Section 2.1, first paragraph line 4: Change “…study was showed in Table2” to “…study is shown in Table 2.”
  • Discussion, line 2: “In present study, we found that 3 RJ fatty acids consisting of C8-10 chains, 10HDA/8HOA/3,10DDA, in combination with ACh are potential RJ components for the treatment of dry eye” should be changed to 10HDA/8HOA/3,10DDA, in combination with ACh are the RJ components that may have potential application in the treatment of dry eye

2.Table 1 and 2 can be combined. Please provide a reference for Ach concentration in RJ in footnotes of Table.

3.Section 2.5: The authors state that “The mice tear secretion was measured using a modified Schirmer test with a phenol red thread (Zone-Quick; Showa Yakuhin Kako, Tokyo, Japan)”. It appears that they have performed the phenol red thread test, which is a well-recognised test on its own. Therefore, it is not right to call it a “modified Schirmer test”. The Schirmer test is much more invasive and measures reflex tear volume whereas, the phenol red thread test, being less invasive measures basal tear volume.

4.The conclusion needs to be written better. While this study wasn’t the first to show Ach is present in RJ, it shows that the presence of Ach is critical for the therapeutic activity of RJ in DED. The authors should also mention the DED parameters in which an improvement was observed (tear volume, t5ear secretion, LG morphology, etc.)

Reviewer 3 Report

The manuscript entitled “Combination of acetylcholine and fatty acids in royal jelly are potential components for the treatment of dry-eye mouse model” is designed well and has the novelty for the application of new combination therapy for DED. However, it needs some edits before publication.

As per the DED mechanism, Ach causes stimulation of secretion and was shown to be regulated by the mitogen activated protein kinase p38, a protein previously not known to be involved in exocrine secretion. RJ itself has Ach as one of the component. How the evaporation type DED treated with RJ.

The animal model, how the mouse model correlated with human lacrimal secretions. What is the rationale for selecting the mouse model?  What is the reason for female mouse only?

As per the literature report, chemical induced or atropine induced methods were more reproducible and rigorous methods for DED induction. How to validate and confirm the stress induced method in DED induction.

The selection of fatty acids, what is the basis for the proposed studies.

The synthesis method and instruments analysis data are provided in the supporting information section. Write the Supply figures in this section and also write a note on how to confirm based on the Fig. S1 and S2.

In the animals section, write the age and weight of the animals used.

How to quantify the contents of each fatty acids from ETRJ, in vitro.

Write the needle or gauge used for the oral administration.

Distilled water was used as a vehicle control. Is this group disease induced or normal mouse group?

Why the disease control group not used for the comparison of the data.

Specify number animals used for each treatment.

Section 2.9 – write the method validation parameters as per the experimental conditions.

Line# 195, delete ‘enzyme-treated RJ’.

Line# 196, ETRJ and raw RJ significantly – write the level of significance.

ETRJ and raw RJ significantly preserved tear secretion, and ACh and 9 RJ FAs. What is the effect of ETRJ and RT on tear secretion compared with Ach, 9 RJ FAs and RJ only?

In figure 1, (A and B) All data represents the mean ± SD, n=6 eyes. n = 6 eyes include both left and right eye from each animal or one eye from six animals. Specify?

Improve the quality of the figures, especially the axis values.

Round 2

Reviewer 3 Report

The manuscript modified as per the comments.